# GANDALF: LEARNING LABEL CORRELATIONS IN EXTREME MULTI-LABEL CLASSIFICATION VIA LABEL FEATURES

## ABSTRACT

Extreme Multi-label Text Classification (XMC) involves learning a classifier that can assign an input with a subset of most relevant labels from millions of label choices. Recent works in this domain have increasingly focused on a symmetric problem setting where both input instances and label features are short-text in nature. Short-text XMC with label features has found numerous applications in areas such as query-to-ad-phrase matching in search ads, title-based product recommendation, prediction of related searches, amongst others. In this paper, we propose *Gandalf*, a novel approach which makes use of a label correlation graph to leverage label features as additional data points to supplement the training distribution. By exploiting the characteristics of the short-text XMC problem, it leverages the label features to construct valid training instances, and uses the label graph for generating the corresponding soft-label targets, hence effectively capturing the label-label correlations. While most recent advances in XMC have been algorithmic, mainly aimed towards developing novel deep-learning frameworks, our data-centric augmentation approach is orthogonal to these methodologies, and can be applied in a plug-and-play manner to a variety of them. This generality and effectiveness of *Gandalf* is demonstrated by showing up to 30% relative improvements for 5 state-of-the-art algorithms across 4 benchmark datasets consisting of up to 1.3 million labels.

## 1    INTRODUCTION

*Extreme Multilabel Classification* (XMC) has found numerous applications in the domains of related searches (Jain et al., 2019), dynamic search advertising (Prabhu et al., 2018) and recommendation tasks such as query-to-ad-phrase (Dahiya et al., 2021b), query-to-product (Medini et al., 2019), product-to-product (Mittal et al., 2021a), etc., which require predicting the most relevant results that frequently co-occur together (Chiang et al., 2019; Hu et al., 2020), or are highly correlated to the given product or search query. These tasks are often modeled through embedding-based retrieval-cum-ranking pipelines over millions of possible web page titles, products titles, or ad-phrase keywords forming the label space. A major challenge in XMC problems is caused by the long-tailed label distribution, i.e., the presence of tail labels with extremely scarce training data. In this paper, we focus on the short-text setting, where we argue there exists a symmetry between inputs and labels, which can be exploited for improved learning of label correlations.

**Extreme class imbalance:** The real world datasets in XMC are highly imbalanced towards popular or trending items. Moreover, these datasets adhere to Zipf's law (Adamic & Huberman, 2002; Ye et al., 2020), i.e., following a long-tailed distribution, where most labels are tail labels with very few ($\leq 5$) positive data-points in a training set spanning $\geq 10^6$ total data points (Table 1). Consequently, the label co-occurrence graph of XMC datasets is extremely sparse, i.e., the presence of a given label does not really imply the presence of other labels (Babbar & Schölkopf, 2019). This makes capturing correlations among labels for the encoder a challenging task. More so, since the encoder is forced to rely solely on the sparse instance-to-label (i.e. input-to-output) mappings, which are often insufficient, especially for tail labels. Such an inherent disconnect in the label co-occurrence graph also motivates utilisation of the one-vs-rest classifier as a popular choice for XMC algorithms.

**Symmetric nature of short-text XMC with Label Features**: Applications of short-text XMC ranging from query-to-ad-phrase prediction to title-based product-to-product recommendation witness short-text instances not only in the input space, but also in the output space. Ad-phrases or product-titles spanning the output space as label descriptors are, like the input query, short-text instances which, on average, consist of only 3-8 words (Dahiya et al., 2021a). Earlier works in XMC primarily focused on problems where the labels were identified by numeric IDs, and hence devoid of any semantic meaning. Here, works (Dahiya et al., 2021b) focused only on learning the nuances of short-text inputs for the XMC task. However, more recently, with inclusion of label descriptors – known as *label features* – in the output space, short-text XMC has taken a symmetric form. This has enabled researchers to more effectively capture the nuances shared between input text and label features in a common embedding space (Mittal et al., 2021a;b; Dahiya et al., 2021a; 2023).

**Learning Label Correlations:** While label correlations are difficult to learn in XMC, previous approaches like (Guo et al., 2019; Mittal et al., 2021b) have tried modelling them in different ways. However, the general idea has been to introduce a label correlation graph (LCG) in the pipeline to implicitly capture higher-order correlations missing in the query's ground-truth. In contrast, we take a unique data-centric approach and propose to leverage the innate symmetry of short-text XMC along the LCG to construct valid data-points. Our proposed approach, which we refer to as *Gandalf*, is a novel method to leverage label features as data points trained through supervisory signals in the form of higher-order correlations from the LCG. The output label vector, when the label features are used as the input data, is constructed by its normalized correlation vector with other labels, that is captured by the LCG. As a consequence, projecting labels from the input (as features) to output (as classification label vectors) space helps the encoder learn representations which are inherently endowed with stronger label correlation information via supervised learning as opposed to simply (i) leveraging the label features for contrastive learning (Dahiya et al., 2021a; 2023) or, (ii) augmenting the classifiers with LCG (Mittal et al., 2021b). To summarise, our contributions are the following:

- We propose *Gandalf* — **G**raph **A**ugme**N**ted **DA**ta with **L**abel **F**eatures — a simple yet effective algorithm to efficiently leverage label features to construct additional training instances based on exploiting the unique setting of short-text XMC in a novel manner.

- In terms of prediction performance, we demonstrate the generality and effectiveness of *Gandalf* by showing up to 30% gains on 5 state-of-the-art extreme classifiers across 4 public benchmarks. We show that by using *Gandalf*, XMC methods which inherently do not leverage label features beat or parallel strong baselines which either employ elaborate training pipelines (Dahiya et al., 2021a), large transformer encoders (You et al., 2019; Zhang et al., 2021b; Dahiya et al., 2023) or make heavy architectural modifications (Mittal et al., 2021a;b) to leverage label features.

- Finally, *Gandalf* does not add any additional computational overhead during inference over the base algorithm. Moreover, unlike other methods which try to capture label correlations (Saini et al., 2021; Chien et al., 2023; Mittal et al., 2021b), *Gandalf* is designed to keep the memory costs constant while learning, only compromising on the training time due to added data points, thus widening its applicability.

## 2 RELATED WORK

**Previous XMC Works**: Prior works in XMC focused on annotating long-text documents, consisting of hundreds of word tokens, such as those encountered in tagging for Wikipedia (Babbar & Schölkopf, 2017; You et al., 2019) with numeric label IDs. Most works under this setting were aimed towards scaling up transformer encoders for XMC task (Zhang et al., 2021b; Kharbanda et al., 2022). With the introduction of label features, there exist three correlations that can be exploited for better representation learning: (i) query-label, (ii) query-query, and (iii) label-label correlations.

**Exploiting Correlations in XMC:** Recent works have been successful in leveraging label features and pushing state-of-the-art by exploiting the first two correlations. For example, SIAMESEXML and NGAME (Dahiya et al., 2021a; 2023) employ a two-tower pre-training stage applying contrastive learning between an input text and its corresponding label features. GALAXC (Saini et al., 2021) & PINA (Chien et al., 2023), motivated by graph convolutional networks, create a combined query-label bipartite graph to aggregate predicted instance neighbourhood. This approach, however, leads to a multifold increase in the memory footprint. DECAF and ECLARE (Mittal et al., 2021a;b)

make architectural additions to embed label-text embeddings (LTE) and graph-augmented label embeddings (GALE) in each label's OVA classifier to exploit higher order correlations from LCG.

**Two-tower Models & Classifier Learning**: Typically, due to the single-annotation nature of most dense retrieval datasets (Nguyen et al., 2016; Kwiatkowski et al., 2019; Joshi et al., 2017), two-tower models (Karpukhin et al., 2020) solving this task eliminate classifiers in favour of modelling implicit correlations by bringing query-document embeddings closer in the latent space of the encoder. These works are conventionally aimed at improving encoder representations by innovating on hard-negative mining (Zhang et al., 2021a; Xiong et al., 2021; Lu et al., 2022), teacher-model distillation (Qu et al., 2021; Ren et al., 2021) and combined dense-sparse training strategies (Khattab & Zaharia, 2020). While these approaches result in enhanced encoders, the multilabel nature of XMC makes them, in itself, insufficient for this domain. This has been demonstrated in two-stage XMC works like (Dahiya et al., 2021a; 2023) where these frameworks go beyond two-tower training and train classifiers with a frozen encoder in the second stage for better empirical performance (Table 2).

## 3 PRELIMINARIES

For training, we have available a multi-label dataset $\mathcal{D} = \{\{\mathbf{x}_i, \mathsf{y}_i\}_{i=1}^N, \{\mathbf{z}_l\}_{l=1}^L\}$[1] comprising of $N$ data points. Each $i \in [N]$ is associated with a small ground truth label set $\mathsf{y}_i \subset [L]$ from $L \sim 10^6$ possible labels. Further, $\mathbf{x}_i, \mathbf{z}_l \in \mathcal{X}$ denote the textual descriptions of the data point $i$ and the label $l$ which, in this setting, derive from the same vocabulary universe $\mathcal{V}$ (Dahiya et al., 2021a). The goal is to learn a parameterized function $f$ which maps each instance $\mathbf{x}_i$ to the vector of its true labels $\mathbf{y}_i \in \{0, 1\}^L$ where $\mathbf{y}_{il} = 1 \Leftrightarrow l \in \mathsf{y}_i$.

A common strategy for handling this learning problem, the *two towers* approach, is to map instances and labels into a common Euclidean space $\mathcal{E} = \mathbb{R}^d$, in which the relevance $s_l(\mathbf{x})$ of a label $l$ to an instance is scored using an inner product, $s_l(\mathbf{x}) = \langle \Phi(\mathbf{x}), \Psi(l) \rangle$. We call $\Phi(\mathbf{x})$ the encoding representation of the instance $\mathbf{x}$, and $\mathbf{w}_l := \Psi(l)$ the decoding representation of label $l$. If labels are featureless integers, then $\Psi$ turns into a simple table lookup. In our setting, $l$ is associated with features $\mathbf{z}_l$, so we identify $\Psi(l) = \Psi(\mathbf{z}_l)$.

The prediction function selects the $k$ highest-scoring labels, $f(\mathbf{x}) = \text{top}_k (\langle \Phi(\mathbf{x}), \Psi(\cdot) \rangle)$. Training is usually handled using the *one-vs-all* paradigm, which applies a binary loss function $\ell$ to each entry in the score vector. In practice, performing the sum over all labels for each instance is prohibitively expensive, so the sum is approximated by a shortlist of labels $\mathsf{S}(\mathbf{x}_i)$ that typically contains all the positive labels, and only those negative labels which are expected to be challenging for classification (Dahiya et al., 2021a;b; 2023; Zhang et al., 2021b; Kharbanda et al., 2023), leading to

$$\mathcal{L}_{\mathcal{D}}[\Phi, \Psi] = \sum_{i=1}^N \sum_{l=1}^L \ell(\mathbf{y}_{il}, \langle \Phi(\mathbf{x}), \Psi(l) \rangle) \approx \sum_{i=1}^N \sum_{l \in \mathsf{S}(\mathbf{x}_i)} \ell(\mathbf{y}_{il}, \langle \Phi(\mathbf{x}), \Psi(l) \rangle). \quad (1)$$

Even though these approaches have been used with success, they still struggle in learning good embeddings $\mathbf{w}_l$ for long-tail labels: A classifier that learns solely based on instance-label pairs has little chance of learning similar label representations for labels that do not co-occur within the dataset, even though they might be semantically related. Consequently, training can easily lead to overfitting even with simple classifiers (Guo et al., 2019).

To reduce the generalization gap, regularization needs to be applied to the label decoder $\Psi$, either explicitly as a new term in the loss function (Guo et al., 2019), or implicitly through the inductive biases of the network structure (Mittal et al., 2021a;b) or by a learning algorithm (Dahiya et al., 2021a; 2023). These approaches incorporate additional label metadata – *label features* – to generate the inductive biases. For short-text XMC, these features themselves are often short textual description, coming from the same space as the instances, as the following examples, taken from (i) LF-AmazonTitles-131K (recommend related products given a product name) and (ii) LF-WikiTitles-500K (predict relevant categories, given the title of a Wikipedia page) illustrate:

*Example 1:* For *"Mario Kart: Double Dash!!"* on Amazon, we have available: *Mario Party 7 | Super Smash Bros Melee | Super Mario Sunshine | Super Mario Strikers* as the recommended products.

---

[1] bold symbols $\mathbf{y}$ indicate vectors, captial letters $Y$ indicate random variables, and sans-serif $\mathsf{y}$ denotes a set

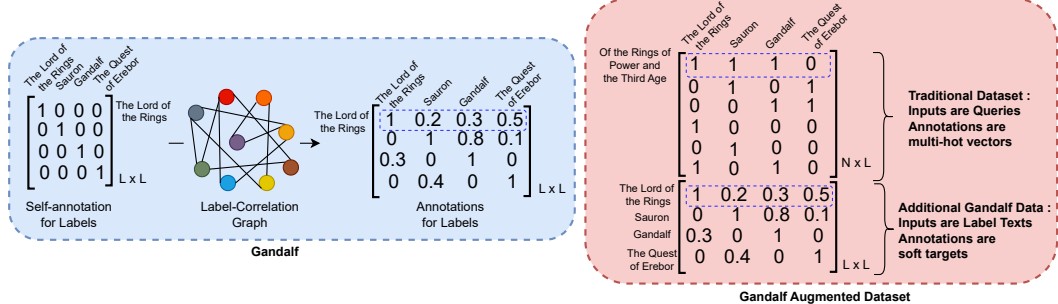

Figure 1: Figure showing how Gandalf augments the training dataset. Soft-targets for each label in the label-space are derived from the label correlation graph. These additional datapoints are simply concatenated with the traditional dataset(that contains queries with hard-targets) for training.

*Example 2:* For the Wikipedia page *"2022 French presidential election"*, we have the available categories: *April 2022 events in France | 2022 French presidential election | 2022 elections in France | Presidential elections in France*. Further, a google search of the same query, leads to the following related searches - *French election 2022 - The Economist | French presidential election coverage on FRANCE 24 | Presidential Election 2022: A Euroclash Between a "Liberal... | French polls, trends and election news for France - POLITICO.eu*, amongst others.

In view of these examples, one can affirm two important observations: (i) the short-text XMC problem indeed requires recommending similar items which are either highly correlated or co-occur frequently with the queried item, and (ii) the queried item and the corresponding label-features form an "equivalence class" and convey similar intent (Dahiya et al., 2021a). For example, a valid news headline search should either result in a page mentioning the same headline or similar headlines from other media outlets (see Example 2). As a result, it can be argued that data instances are *interchangeable* with their respective labels' features.

Exploiting this interchangeability of label and instance text, Dahiya et al. (2021a; 2023) proposes to tie encoder and decoder together and require $\Psi(l) = \Phi(\mathbf{z}_l)$. While indeed yielding improved test performance, this approach has two drawbacks: Firstly, the condition $\Psi(l) = \Phi(\mathbf{z}_l)$ turns out to be too strong, and it has to allow for some fine-tuning corrections $\boldsymbol{\eta}_l$, yielding $\Psi(l) = \Phi(\mathbf{z}_l) + \boldsymbol{\eta}_l$. Consequently, training of SIAMESEXML and NGAME is done in two stages: Initially, a contrastive loss needs to be minimized, followed by fine-tuning with a classification objective.

## 4 GANDALF: LEARNING LABEL-LABEL CORRELATIONS

Dahiya et al. (2021a) motivate their approach by postulating a self-annotation property (*Label Self Proximity*), which claims that a label $l$ is relevant to its own textual features with high probability, $\mathbb{P}[Y_l = 1 \mid X = \mathbf{z}_l] > 1 - \epsilon$ for some small $\epsilon \ll 1$. Another recent work Chien et al. (2023), in it's pretraining step, attempts to leverage label features as data points but does so by expanding the label space $\{0,1\}^L$ to also include instances as $\{0,1\}^{L+N}$ leveraging the self-annotation property of labels and inverting the initial instance-label mappings to have instances $\mathbf{x}_i$ as labels for label features $\mathbf{z}_l$ as data points. This, however, leads to an explosion in an already enormous label space.

Most instances in an XMC problem are associated with multiple labels, yet the self-annotation postulate only provides a single label. Thus, one might ask, *Question: In a label space spanning the order of $10^6$ labels, what are the other labels which annotate $\mathbf{z}_l$, when posed as a data point?* In order to effectively augment the training set with $\mathbf{z}_l$ as a data point, we need to provide values for the other entries of the label vector $\mathbf{y}_l$. Ideally, these labels would be sampled according to $\mathbf{y}_l \sim \mathbb{P}[\mathbf{Y} \mid X = \mathbf{z}_l]$, which means we need to find sensible approximations to the probabilities for the other labels $\mathbb{P}[Y_j = 1 \mid X = \mathbf{z}_l]$. When using the cross-entropy loss, sampling can be forgone by replacing the discrete labels $\mathbf{y}_l \in \{0,1\}^L$ by soft labels $\mathbf{y}_l^{\text{soft}} = \mathbb{P}[\mathbf{Y} \mid X = \mathbf{z}_l]$.

In order to derive a model for $\mathbb{P}[Y_{l'} = 1 \mid X = \mathbf{z}_l]$, we can take inspiration from the GLAS regularizer (Guo et al., 2019). This regularizer tries to make the Gram matrix of the label embeddings $\langle \mathbf{w}_l, \mathbf{w}_{l'} \rangle$ reproduce the co-occurrence statistics of the labels $\mathbf{S}$,

$$\mathcal{R}_{\text{GLaS}}[\Psi] = L^{-2} \sum_{l=1}^{L} \sum_{l'=1}^{L} \left( \langle \mathbf{w}_l, \mathbf{w}_{l'} \rangle - S_{ll'} \right)^2. \tag{2}$$

Here, $\mathbf{S}$ denotes the symmetrized conditional probabilities,

$$S_{ll'} := 0.5(\mathbb{P}[Y_l = 1 \mid Y_{l'} = 1] + \mathbb{P}[Y_{l'} = 1 \mid Y_l = 1]). \tag{3}$$

Plugging in $\mathbf{w}_l = \Psi(\mathbf{z}_l)$, this regularizer reaches its minimum if

$$\langle \Psi(\mathbf{z}_l), \Psi(\mathbf{z}_{l'}) \rangle = S_{ll'}. \tag{4}$$

By the self-proximity postulate, we can assume $\Psi(\mathbf{z}_l) \approx \Phi(\mathbf{z}_l)$. For a given label feature instance with target soft-label $(\mathbf{z}_l, y_{ll'}^{\text{soft}})$, the training will try to minimize $\ell(\langle \Phi(\mathbf{z}_l), \Psi(\mathbf{z}_{l'}) \rangle, y_{ll'}^{\text{soft}})$. To be consistent with Equation 4, we therefore want to choose $y_{ll'}^{\text{soft}}$ such that $S_{ll'} = \arg\min \ell(\cdot, y_{ll'}^{\text{soft}})$. This is fulfilled for $y_{ll'}^{\text{soft}} = \sigma(S_{ll'})$ for $\ell$ being the binary cross-entropy, where $\sigma$ denotes the logistic function. If $\ell$ is the squared error, then the solution is even simpler, with $y_{ll'}^{\text{soft}} = S_{ll'}$.

For simplicity, and because of strong empirical performance, we choose $y_{ll'}^{\text{soft}} = S_{ll'}$ even when training with cross-entropy loss. This results in the extended version of the self-proximity postulate:

**Postulate 1 (Soft-Labels for Label Features)** *Given a label $l$ with features $\mathbf{z}_l \in \mathcal{X}$, and a proxy for semantic similarity of labels $\mathbf{S}$, the labels features, when interpreted as an input instance, should result in predictions*

$$\mathbb{P}[Y_{l'} = 1 \mid X = \mathbf{z}_l] \approx S_{ll'}. \tag{5}$$

More intuitively, Postulate 1 answers the *Question* posed above by stating that the label vector $\mathbf{y}$, when a given label feature $\mathbf{z}_l$ is used as an instance, should have the soft-label value $y_{ll'}^{\text{soft}}$ that is in proportion to the $l'$ entry in the LCG corresponding for the given $l$.

The label-similarity measure (Equation 3) used in the original GLAS regularizer uses only direct co-occurences of labels, which results in a noisy signal that does not capture higher-order label interdependencies. In contrast, the LCG $\in \mathbb{R}^{L \times L}$ (Mittal et al., 2021b) is inferred by performing a random walk (with restarts) over the bipartite graph connecting input data instances with their corresponding ground-truth labels. Since the entries in the LCG are normalized and skewed in favor of tail labels, the LCG can be interpreted as a smoothed and regularized variant of the label co-occurrence matrix. This enables LCG to correctly identify a set of semantically similar labels that either share tokens with the queried label, or co-occur frequently in the same context. We show this qualitatively in Appendix B, where the "History of Computing" is the most similar to "Computer Museums" and "Charles Babbage Institute". This property makes its edge weights ($\mathcal{G}_{i,j}$) a good candidate for the similarty measure $S_{ij}$.

While originally LCG was utilized to efficiently mine higher order query tail-label relations by augmenting the classifier $\Psi$ with graph information, we propose to leverage the graph weights (with an additional row-wise normalization to get values in range [0, 1]) as probabilistic soft labels for $\mathbf{z}_l$ as data instance. Further, to restrict the impact of noisy correlations in large output spaces Babbar & Schölkopf (2019), we empirically find it beneficial to threshold the soft labels obtained from LCG at $\delta$. Thus, we propose a novel method - *Gandalf* - to augment the training dataset with label features as additional data points, annotated by label vectors given by:

$$\mathbf{y}_{ij} = \mathbb{P}[Y_i = 1 \mid X = \mathbf{z}_j] \approx S_{ij} = \begin{cases} \mathcal{G}_{ij}/\mathcal{G}_{jj}, & \mathcal{G}_{ij} > \delta \\ 0, & \mathcal{G}_{ij} \leq \delta \end{cases} \tag{6}$$

A diagrammatic representation of our method is provided in Figure 1. We hypothesize that the models benefit from *Gandalf* in two ways. First, *Gandalf* leverages label features, existing in the same distribution and vocabulary as $\mathcal{D}$ (Dahiya et al., 2021a), as novel data points which are then used for training in a supervised setting to mimic the apriori statistical correlations between labels that exist in the label space. The model is able to capture these correlations because *Gandalf* is multilabel in nature and forces the encoder to learn to create a single representation for the label features that

maximizes the probability score across all positive labels (OVA classifiers). Secondly, as shown in section 3, data points more-often-than-not share tokens with their label's features. Therefore training models with these additional data points helps better learn both input token embeddings and classifier label embeddings i.e. OVA decision boundaries.

As a result, the encoded representation of correlated labels, learnt by an underlying algorithm, are closer in the representation space. This especially benefits the tail labels which, more often than not, either get missed out during shortlisting or rank outside the desired top-k predictions.

## 5 EXPERIMENTS & DISCUSSION

| Datasets | N | L | APpL | ALpP | AWpP |
|---|---|---|---|---|---|
| LF-AmazonTitles-131K | 294,805 | 131,073 | 5.15 | 2.29 | 6.92 |
| LF-WikiSeeAlsoTitles-320K | 693,082 | 312,330 | 4.67 | 2.11 | 3.01 |
| LF-WikiTitles-500K | 1,813,391 | 501,070 | 17.15 | 4.74 | 3.10 |
| LF-AmazonTitles-1.3M | 2,248,619 | 1,305,265 | 38.24 | 22.20 | 8.74 |

Table 1: Details of short-text benchmark datasets with label features. APpL stands for avg. points per label, ALpP stands for avg. labels per point and AWpP is the length i.e. avg. words per point.

**Benchmarks, Baselines & Metrics** We benchmark our experiments on 4 standard public datasets, the details of which are mentioned in Table 1. To test the generality and effectiveness of our proposed *Gandalf*, we apply the algorithm across multiple state-of-the-art short-text extreme classifiers: (i) ASTEC, (ii) DECAF, (iii) ECLARE, and (iv) INCEPTIONXML. Furthermore, we also compare against two-tower approaches like DPR (Karpukhin et al., 2020), ANCE (Xiong et al., 2021), RocketQA (Qu et al., 2021), including XMC-specific ones - SIAMESEXML and NGAME. We do not evaluate *Gandalf* on two-tower approaches since it is non-trivial and part of our future work to implement it with a two-tower training objective. Additionally, we also compare against transformer-encoder based AttentionXML (You et al., 2019) and XR-Transformer (Zhang et al., 2021b).

As an algorithmic contribution, we extend the INCEPTIONXML encoder to leverage label features to further the state-of-the-art on benchmark datasets and call it INCEPTIONXML-LF. For this, we augment the OVA classifier with additional label-text embeddings (LTE) and graph-augmented label embeddings (GALE), as done in Mittal et al. (2021b). The implementation details and training strategy can be found in Appendix A. We measure the performance using standard metrics P@k, and its propensity-scored variant, PSP@k (Jain et al., 2016).

### 5.1 EMPIRICAL PERFORMANCE

With *Gandalf*, gains of up to 30% can be observed in case of ASTEC and INCEPTIONXML which, by default, do not leverage label features and yet perform at par and sometimes better as compared their LF-counterparts, i.e. DECAF and ECLARE, and INCEPTIONXML-LF across all datasets. DECAF and ECLARE tie the embedding layer with one (LTE) or two (LTE + GALE) additional ASTEC-like encoders respectively to take advantage of label text embeddings (LTE) and graph augmented label embeddings (GALE) in order to feed label feature information in classifiers. While architectural modifications help capture higher order query-label relations and help model predict unseen labels better, they add significant computational overhead. DECAF (having LTE) is $\sim 2\times$ expensive to train as compared to its base model ASTEC while ECLARE (having both LTE & GALE) adds up to $\sim 3\times$ computational cost. Similar trend is witnessed between INCEPTIONXML and its modified LF counterpart. On the other hand, base encoders trained with *Gandalf*, imbue necessary correlations without needing to make any additional modifications or employ complicated training pipelines.

It may be noted that all 5 encoders trained with additional *Gandalf*-generated data points are frugal architectures trained from scratch. While ASTEC merely consists of a linear layer as an encoder and makes use of an ANNS for prediction, INCEPTIONXML employs only a self-attention layer followed by two convolutional and two linear layers. On the other hand, two tower approaches, except SIAMESEXML – which uses an ASTEC encoder, employ pre-trained DistilBERT or BERT models. Our findings from Table 2 are in-line with those from Kharbanda et al. (2023) where they show that transformer models are perhaps an overkill for the short-text XMC task at hand. Frugal

| Method | P@1 | P@3 | P@5 | PSP@1 | PSP@3 | PSP@5 | P@1 | P@3 | P@5 | PSP@1 | PSP@3 | PSP@5 |
|---|---|---|---|---|---|---|---|---|---|---|---|---|
| | | | LF-AmazonTitles-131K | | | | | | LF-AmazonTitles-1.3M | | | |
| AttentionXML | 32.25 | 21.70 | 15.61 | 23.97 | 28.60 | 32.57 | 45.04 | 39.71 | 36.25 | 15.97 | 19.90 | 22.54 |
| DPR* | 41.85 | 28.71 | 20.88 | 38.17 | 43.93 | 49.45 | 44.64 | 39.05 | 34.83 | 32.62 | 35.37 | 36.72 |
| ANCE* | 42.67 | 29.05 | 20.98 | 38.16 | 43.78 | 49.03 | 46.44 | 41.48 | 37.59 | 31.91 | 35.31 | 37.25 |
| ROCKETQA* | 42.75 | 29.22 | 20.98 | **39.97** | **44.50** | **49.21** | - | - | - | - | - | - |
| XR-TRANSFORMER | 38.10 | 25.57 | 18.32 | 28.86 | 34.85 | 39.59 | 50.14 | 44.07 | 39.98 | 20.06 | 24.85 | 27.79 |
| NGAME* | 42.61 | 28.86 | 20.69 | 38.27 | 43.75 | 48.71 | 45.82 | 39.94 | 35.48 | **33.03** | **35.63** | **36.80** |
| + classifier | **44.95** | **29.87** | **21.20** | - | - | - | **54.69** | **47.08** | **42.80** | - | - | - |
| SIAMESEXML* | 38.36 | 26.20 | 19.26 | 34.83 | 39.87 | 45.18 | - | - | - | - | - | - |
| + classifier | 41.42 | **30.19** | 21.21 | 35.80 | 40.96 | 46.19 | 49.02 | 42.72 | 38.52 | 27.12 | 30.43 | 32.52 |
| ASTEC | 37.12 | 25.20 | 18.24 | 29.22 | 34.64 | 39.49 | 48.82 | 42.62 | 38.44 | 21.47 | 25.41 | 27.86 |
| + Gandalf | 43.95 | 29.66 | 21.39 | 37.40 | 43.03 | 48.31 | 53.02 | 46.13 | 41.37 | 27.32 | 31.20 | 33.34 |
| DECAF | 38.40 | 25.84 | 18.65 | 30.85 | 36.44 | 41.42 | 50.67 | 44.49 | 40.35 | 22.07 | 26.54 | 29.30 |
| + Gandalf | 42.43 | 28.96 | 20.90 | 35.22 | 42.12 | 47.61 | 53.02 | 46.65 | 42.25 | 25.47 | 30.14 | 32.83 |
| ECLARE | 40.46 | 27.54 | 19.63 | 33.18 | 39.55 | 44.10 | 50.14 | 44.09 | 40.00 | 23.43 | 27.90 | 30.56 |
| + Gandalf | 42.51 | 28.89 | 20.81 | 35.72 | 42.19 | 47.46 | **53.87** | **47.45** | **43.00** | 28.86 | 32.90 | 35.20 |
| INCEPTIONXML | 36.79 | 24.94 | 17.95 | 28.50 | 34.15 | 38.79 | 48.21 | 42.47 | 38.59 | 20.72 | 24.94 | 27.52 |
| + Gandalf | **44.67** | 30.00 | **21.50** | 37.98 | 43.83 | 48.93 | 50.80 | 44.54 | 40.25 | 25.49 | 29.42 | 31.59 |
| INCEPTIONXML-LF | 40.74 | 27.24 | 19.57 | 34.52 | 39.40 | 44.13 | 49.01 | 42.97 | 39.46 | 24.56 | 28.37 | 31.67 |
| + Gandalf | 43.84 | 29.59 | 21.30 | **38.22** | **43.90** | **49.03** | 52.91 | 47.23 | 42.84 | **30.02** | **33.18** | **35.56** |
| | | | LF-WikiSeeAlsoTitles-320K | | | | | | LF-WikiTitles-500K | | | |
| AttentionXML | 17.56 | 11.34 | 8.52 | 9.45 | 10.63 | 11.73 | 40.90 | 21.55 | 15.05 | 14.80 | 13.97 | 13.88 |
| SIAMESEXML++ | 31.97 | 21.43 | 16.24 | **26.82** | 28.42 | 30.36 | 42.08 | 22.80 | 16.01 | 23.53 | 21.64 | 21.41 |
| ASTEC | 22.72 | 15.12 | 11.43 | 13.69 | 15.81 | 17.50 | 44.40 | 24.69 | 17.49 | 18.31 | 18.25 | 18.56 |
| + Gandalf | 31.10 | 21.54 | 16.53 | 23.60 | 26.48 | 28.80 | 45.24 | 25.45 | 18.57 | 21.72 | 20.99 | 21.16 |
| DECAF | 25.14 | 16.90 | 12.86 | 16.73 | 18.99 | 21.01 | 44.21 | 24.64 | 17.36 | 19.29 | 19.82 | 19.96 |
| + Gandalf | 31.10 | 21.60 | 16.31 | 24.83 | 27.18 | 29.29 | 45.27 | 25.09 | 17.67 | 22.51 | 21.63 | 21.43 |
| ECLARE | 29.35 | 19.83 | 15.05 | 22.01 | 24.23 | 26.27 | 44.36 | 24.29 | 16.91 | 21.58 | 20.39 | 19.84 |
| + Gandalf | 31.33 | 21.40 | 16.31 | 24.83 | 27.18 | 29.29 | 45.12 | 24.45 | 17.05 | 24.22 | 21.41 | 20.55 |
| INCEPTIONXML | 23.10 | 15.54 | 11.52 | 14.15 | 16.71 | 17.39 | 44.61 | 24.79 | 19.52 | 18.65 | 18.70 | 18.94 |
| + Gandalf | 32.54 | 22.15 | 16.86 | 25.27 | 27.76 | 30.03 | 45.93 | 25.81 | **20.36** | 21.89 | 21.54 | 22.56 |
| INCEPTIONXML-LF | 28.99 | 19.53 | 14.79 | 21.45 | 23.65 | 25.65 | 44.89 | 25.71 | 18.23 | 23.88 | 22.58 | 22.50 |
| + Gandalf | **33.12** | **22.70** | **17.29** | 26.68 | **29.03** | **31.27** | **47.13** | **26.87** | 19.03 | **24.12** | **23.92** | **23.82** |

Table 2: Results showing the effectiveness of *Gandalf* on state-of-the-art extreme classifiers. (*) denotes two-tower models. The best-performing approach are put in **bold**. For Amazon datasets, best results using frugal architectures and those using transformers(highlighted) are **bold** separately.

architectures, which by themselves, under performed as compared to two-tower approaches, parallel or surpass the same when trained with *Gandalf* augmented data.

## 5.2 DISCUSSION

We can make some key observations and develop strong insights not only about the short-text XMC problem with label features but also about specific dataset properties from Table 2. For example, training on label features as data points generated via *Gandalf* gives remarkable improvements on top of existing algorithms, especially on LF-AmazonTitles-131K and LF-WikiSeeAlsoTitles-320K where most labels have ~5 training data points on average. In these low data regimes, *Gandalf* helps imbue important label correlation information in the encoder which is missed by most training algorithms employed by existing models. In contrast, improvements on LF-WikiTitles-500K remain relatively mild, where there is enough data per label for the existing model training algorithms to be able to learn inherent label correlation information by existing query-label mappings.

***Gandalf* vs ECLARE** ECLARE leverages LCG to encode higher-order document-label correlations wherease *Gandalf* explicitly aims to learn label-label correlations. More specifically, in ECLARE, the loss participants are $\mathcal{L}(\phi(\mathbf{x}_i)$ , $\psi + \phi_{LTE}(\mathbf{z}_l) + \phi_{GALE}(\mathbf{z}_l))$ whereas in *Gandalf*, they are $\mathcal{L}(\phi(\mathbf{z}_l), \psi)$. Notably, the correlations learned by *Gandalf* are independent of those captured by GALE in ECLARE. Thus, we find ECLARE and INCEPTIONXML-LF, both of which employ GALE, to benefit off training on data points generated using *Gandalf*. as is evident from the significant improvement in results in Table 2. Further, *Gandalf* simplifies ECLARE's approach and does not need to employ the additional $\phi_{LTE}, \phi_{GALE}$ encoders.

**Two-tower Approaches** As mentioned before, two-tower approaches without classifiers are insufficient for XMC due to the multi-label nature of the problem. This can be clearly seen by the limited performance of state-of-the-art two-tower approaches like DPR (Karpukhin et al., 2020), ANCE (Xiong et al., 2021) and ROCKETQA Qu et al. (2021). Even though RocketQA's elaborate training pipeline consists of cross-encoder teacher-model distillation and data augmentation, it performs only marginally better as compared to NGAME's two-tower training approach. Notably, while two-tower approaches in XMC (Dahiya et al., 2023; 2021a) parallel these dense retrieval methods, they still benefit from the addition of discriminative classifier training. Hence, classifier training is a crucial aspect of XMC pipelines which benefits from the additional supervised signals provided by *Gandalf*.

## 5.3 ABLATIONS AND QUALITATIVE RESULTS

| Method | P@1 | P@3 | P@5 | PSP@1 | PSP@3 | PSP@5 | P@1 | P@3 | P@5 | PSP@1 | PSP@3 | PSP@5 |
|---|---|---|---|---|---|---|---|---|---|---|---|---|
| | | | LF-AmazonTitles-131K | | | | | | LF-WikiSeeAlsoTitles-320K | | | |
| InceptionXML | 35.62 | 24.13 | 17.35 | 27.53 | 33.06 | 37.50 | 21.53 | 14.19 | 10.66 | 13.06 | 14.87 | 16.33 |
| *+ Gandalf w/o SL* | 37.59 | 25.25 | 18.18 | 30.75 | 35.54 | 40.06 | 24.43 | 16.16 | 12.15 | 16.89 | 18.45 | 20.02 |
| *+ Gandalf* | **43.52** | **29.23** | **20.92** | **36.96** | **42.71** | **47.64** | **31.31** | **21.38** | **16.22** | **24.31** | **26.79** | **28.83** |
| INCEPTIONXML-LF | 40.74 | 27.24 | 19.57 | 34.52 | 39.40 | 44.13 | 49.01 | 42.97 | 39.46 | 24.56 | 28.37 | 31.67 |
| *+ Gandalf* ($\delta = 0.$) | 41.71 | 28.03 | 20.14 | 36.94 | 41.93 | 46.64 | 31.40 | 21.56 | 16.53 | 26.01 | 27.89 | 29.99 |
| *+ Gandalf* ($\delta = 0.1$) | **42.09** | **28.38** | **20.45** | **37.09** | **42.19** | **47.04** | **32.20** | **21.86** | **16.60** | **26.06** | **28.01** | **30.03** |
| *+ Gandalf* ($\delta = 0.2$) | 41.73 | 28.10 | 20.18 | 37.01 | 41.99 | 46.67 | 31.29 | 21.35 | 16.28 | 25.68 | 27.59 | 29.65 |
| *+ Gandalf* ($\delta = 0.3$) | 41.39 | 27.74 | 19.89 | 36.71 | 41.51 | 46.09 | 31.03 | 20.92 | 15.99 | 25.11 | 27.12 | 29.14 |

Table 3: Results demonstrating the effectiveness of leveraging label features as data points annotated by soft-labels (denoted *SL*) i.e. *Gandalf* on a single InceptionXML model. The table also shows the method's sensitivity to $\delta$, as defined in Equation 6. Notably, soft-labels are a central aspect in imbuing the encoder with stronger label correlation information.

**Effect of Soft Labels & Sentitivity to $\delta$:** We examine *Gandalf*'s performance without soft-labels on INCEPTIONXML Table 3, where *Gandalf w/o SL* is essentially equivalent to leveraging label features as data points but with self-annotation property alone. However, that only helps the model learn token-to-label associations, like LTE in DECAF. Notably, soft-targets play an important role in enabling the encoder to intrinsically learn the label-label correlations and imbue the necessary inductive bias in the models. We further examine *Gandalf*'s sensitivity to $\delta$ by training INCEPTIONXML-LF on data generated with varying values of $\delta$ on two datasets. As shown in Table 3, the empirical performance peaks at a $\delta$ value of 0.1 which is sufficient to suppresses the impact of noisy correlations. Higher values of $\delta$ tend to suppress useful information from the LCG.

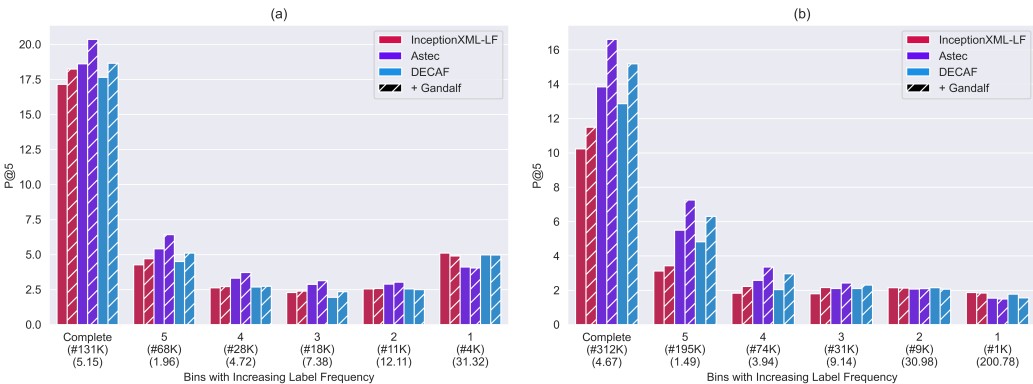

Figure 2: Contributions to P@5 in (a) LF-AmazonTitles-131K and (b) LF-WikiSeeAlsoTitles-320K. The number of labels in each bin is provided after the # in the second row of the tags on the x-axis. The bottommost row denotes the mean label frequency in that bin. Specifically, note the improvements on tail labels in the earlier bins (5 - 3).

**Improvements on tail labels:** We perform a quantile analysis (Figure 2) across 2 datasets – LF-AmazonTitles-131K and the LF-WikiSeeAlsoTitles-320K – where we examine performance (con-

tribution to P@5 metric) over 5 equi-voluminous bins based on increasing order of mean label frequency in the training dataset. Consequently, performance on head labels can be captured by the bin #1 and that of tail labels by bin #5. We note that introducing the additional training data with Gandalf consistently improves the performance across all label frequencies, with more profound gains on bins with more tail labels.

| Method | Datapoint | Baseline Predictions | *Gandalf* Predictions |
|---|---|---|---|
| INCEPTIONXML-LF | | Pontryagin duality, Topological order, Topological quantum field theory, Topological quantum number, Quantum topology | Compact group, Haar measure, Lie group, Algebraic group, Topological ring |
| DECAF | Topological group | Topological quantum computer, Topological order, Topological quantum field theory, Topological quantum number, Quantum topology | Compact group, Haar measure, Lie group, Algebraic group, Topological ring |
| ECLARE | | Topological quantum computer, Topological order, Topological quantum field theory, Topological quantum number, Quantum topology | Compact group, Topological order, Lie group, Algebraic group, Topological ring |
| INCEPTIONXML-LF | | List of lighthouses in Scotland, List of Northern Lighthouse Board lighthouses, Oatcake, Communes of the Finistere department, Communes of the Cotes-d'Armor department | Oatcake, Oatmeal, Oat milk, Porridge, Rolled oats |
| DECAF | Oat | Oatcake, Oatmeal, Design for All (in ICT), Oatley Point Reserve, Oatley Pleasure Grounds | Oatcake, Oatmeal, Oat milk, Porridge, Rolled oats |
| ECLARE | | Oatmeal, Oat milk, Parks in Sydney, Oatley Point Reserve, Oatley Pleasure Grounds | Oatcake, Porridge, Rolled oats, Oatley Point Reserve, Oatley Pleasure Grounds |
| INCEPTIONXML-LF | | Lunar Orbiter Image Recovery Project, Lunar Orbiter 3, Lunar Orbiter 5, Chinese Lunar Exploration Program, List of future lunar missions | Surveyor program, Luna programme, Lunar Orbiter Image Recovery Project, Lunar Orbiter 3, Lunar Orbiter 5 |
| DECAF | Lunar Orbiter program | Exploration of the Moon, List of man-made objects on the Moon, Lunar Orbiter Image Recovery Project, Lunar Orbiter 3, Lunar Orbiter 5 | Exploration of the Moon, Apollo program, Surveyor program, Luna programme, Lunar Orbiter program |
| ECLARE | | Exploration of the Moon, Lunar Orbiter program, Lunar Orbiter Image Recovery Project, Lunar Orbiter 3, Lunar Orbiter 5 | Exploration of the Moon, Pioneer program, Surveyor program, Luna programme, Lunar Orbiter program |

Table 4: Qualitative predictions from the LF-WSAT-320K dataset. Labels indicate mispredictions.

**Qualitative Results:** We further analyse qualitative examples via the top 5 predictions obtained by training the base encoders with and without *Gandalf* augmented data points in Table 4. Additional examples are provided in Appendix B. We note that the quality of predictions increases when the encoder is trained with *Gandalf*-generated data points. For the query "Topological group", where all baselines fail to produce a single correct prediction in top 5, encoders trained with *Gandalf* results in 4/5 correct predictions. Queries with even just a single word, like *"Oat"*, which predicts unrelated labels in the case of the baseline prediction, gets all the labels right with the addition of *Gandalf*. Furthermore, even the quality of incorrect predictions improves i.e. relevance to the input data point increases. For example, in case of "Lunar Orbiter program", the only incorrect *Gandalf* predictions are "Lunar Orbiter 3", "Lunar Orbiter 5" and "Pioneer program" (US lunar and planetary space probes exploration program) which are potential false negatives.

## 6 CONCLUSION

In this paper, we proposed *Gandalf*, a strategy to learn label correlations, a notoriously difficult challenge. In contrast to previous works which model these correlations implicitly through model training, we propose a supervised approach to explicitly learn them by leveraging the inherent query-label symmetry in short-text extreme classification. We further performed extensive experimentation by implementing on various SOTA XMC methods and demonstrated dramatic increases in prediction performances uniformly across all methods. Moreover, this is achieved with frugal architectures without incurring any computational overheads in inference latency or training memory footprint. We hope our treatment of label correlations in this domain will spur further research towards crafting data-points with more expressive annotations, and further extend it to long-text XMC approaches where the instance-label symmetry is quite ambiguous. Learning label correlations in contrastive settings, instead of discriminative ones, as done in two-tower approaches is part of a future work.

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
