# A  INCEPTIONXML-LF

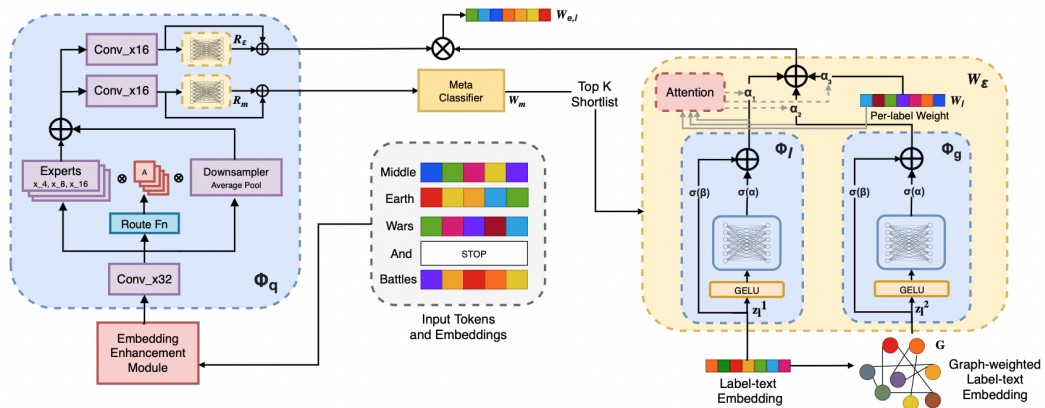

Figure 3: INCEPTIONXML-LF. The improved Inception Module along with instance attention is shown in detail. Changes to the INCEPTIONXML framework using the ECLARE classifier are also shown.

**Model Outlook:** Short-text queries are encoded by a modified InceptionXML encoder, which encodes an input query $\mathbf{x}_i$ using an encoder $\Phi_q := (E, \theta)$ parameterised by $E$ and $\theta$, where $E$ denotes a $D$-dimensional embedding layer of $\mathbb{R}^{\mathcal{V} \times D}$ for vocabulary tokens $\mathcal{V} = [t_1, t_2, \dots, t_V]$ and $\theta$ denotes the parameters of the embedding enhancement and the inception module respectively. Alongside $\Phi_q$, INCEPTIONXML-LF learns two frugal ASTEC-like Dahiya et al. (2021b) encoders, one each as a label-text encoder $\Phi_l := \{E, \mathcal{R}\}$ and a graph augmented encoder $\Phi_g := \{E, \mathcal{R}\}$. Here, $\mathcal{R}$ denotes the parameters of a fully connected layer bounded by a spectral norm and the embedding layer $E$ is shared between all $\Phi_q, \Phi_l$ and $\Phi_g$ for joint query-label word embedding learning. Further, an attention module $\mathcal{A}$, meta-classifier $\mathcal{W}_m$ and an extreme classifier $\mathcal{W}_e$ are also learnt together with the encoders. Next, we specify the details of all components of INCEPTIONXML-LF.

## A.1  INSTANCE-ATTENTION IN QUERY ENCODER

We make two improvements to the inception module INCEPTIONXML for better efficiency. Firstly, in the inception module, the activation maps from the first convolution layer are concatenated before passing them onto the second convolution layer. To make this more computationally efficient, we replace this "inception-like" setting with a "mixture of expert" setting Yang et al. (2019). Specifically, a route function is added that produces dynamic weights for each instance to perform a dynamic element-wise weighted sum of activation maps of each filter. Along with the three convolutional experts, we also add an average pool as a down sampling residual connection to ensure better gradient flow across the encoder. Second, we decouple the second convolution layer to have one each for the meta and extreme classification tasks.

## A.2  DYNAMIC HARD NEGATIVE MINING

Training one-vs-all (OvA) label classifiers becomes infeasible in the XMC setting where we have hundreds of thousands or even millions of labels. To mitigate this problem, the final prediction or loss calculation is done on a shortlist of size $\sqrt{L}$ comprising of only hard-negatives label. This mechanism helps reduce complexity of XMC from an intractable $O(NDL)$ to a computationally feasible $O(ND\sqrt{L})$ problem. INCEPTIONXML-LF inherits the synchronized hard negative mining framework as used in the INCEPTIONXML. Specifically, the encoded meta representation is passed through the meta-classifier which predicts the top-K relevant label clusters per input query. All labels present in the top-K shortlisted label clusters then form the hard negative label shortlist for the extreme task. This allows for progressively harder labels to get shortlisted per short-text query as the training proceeds and the encoder learns better representations.

### A.3 Label-text and LCG Augmented Classifiers

INCEPTIONXML-LF's extreme classifier weight vectors $\mathcal{W}_e$ comprise of 3 weights, as in Mittal et al. (2021b). Specifically, the weight vectors are a result of an attention-based sum of (i) label-text embeddings, created through $\Phi_l$, (ii) graph augmented label embeddings, created through graph encoder $\Phi_g$ and, (iii) randomly initialized per-label independent weights $\mathbf{w}_l$.

As shown in Figure 3, we first obtain label-text embeddings as $\mathbf{z}_l^1 = E \cdot \mathbf{z}_l^0$, where $\mathbf{z}_l^0$ are the TF-IDF weights of label feature corresponding to label $l$. Next, we use the label correlation graph $\mathbf{G}$ to create the graph-weighted label-text embeddings $\mathbf{z}_l^2 = \sum_{m \in [L]} \mathbf{G}_{lm} \cdot \mathbf{z}_l^0$ to capture higher order query-tail label correlations. $\mathbf{z}_l^1$ and $\mathbf{z}_l^2$ are then passed into the frugal encoders $\Phi_l$ and $\Phi_g$ respectively. These encoders comprise only of a residual connection across a fully connected layer as $\alpha \cdot \mathcal{R} \cdot \mathcal{G}(\tilde{z}_l) + \beta \cdot \tilde{z}_l$, where $\tilde{z}_l = \{\mathbf{z}_l^1, \mathbf{z}_l^2\}$, $\mathcal{G}$ represents GELU activation and $\alpha$ and $\beta$ are learned weights. Finally, the per-label weight vectors for the extreme task are obtained as

$$\mathcal{W}_{e,l} = \mathcal{A}(\mathbf{z}_l^1, \mathbf{z}_l^2, \mathbf{w}_l) = \alpha^1 \cdot \mathbf{z}_l^1 + \alpha^2 \cdot \mathbf{z}_l^2 + \alpha^3 \cdot \mathbf{w}_l$$

where $\mathcal{A}$ is the attention block and $\alpha^{\{1,2,3\}}$ are the dynamic attention weights produced by the attention block.

### A.4 Two-phased Training

**Motivation:** We find there to be a mismatch in the training objectives in DeepXML-based approaches like ASTEC, DECAF and ECLARE which first train their word embeddings on meta-labels in Phase I and then transfer these learnt embeddings for classification over extreme fine-grained labels in Phase III Dahiya et al. (2021b). Thus, in our two-phased training for INCEPTIONXML-LF, we keep our training objective same for both phases. Note that, in INCEPTIONXML-LF the word embeddings are always learnt on labels instead of meta-labels or label clusters and we only augment our extreme classifier weight vectors $\mathcal{W}_e$ with label-text embeddings and LCG weighted label embeddings. We keep the meta-classifier $\mathcal{W}_m$ as a standard randomly initialized classification layer.

**Phase I:** In the first phase, we initialize the embedding layer $E$ with pre-trained GloVe embeddings Pennington et al. (2014), the residual layer $\mathcal{R}$ in $\Phi_l$ and $\Phi_g$ is initialized to identity and the rest of the model comprising of $\Phi_q$, $\mathcal{W}_m$ and $\mathcal{A}$ is randomly initialized. The model is then trained end-to-end but without using free weight vectors $\mathbf{w}_l$ in the extreme classifier $\mathcal{W}_e$. This set up implies that $\mathcal{W}_e$ only consists of weights tied to $E$ through $\Phi_l$ and $\Phi_g$ which allows for efficient joint learning of query-label word embeddings Mittal et al. (2021a) in the absence of free weight vectors. Model training in this phase follows the INCEPTIONXML+ pipeline as described in Kharbanda et al. (2023) without detaching any gradients to the extreme classifier for the first few epochs. In this phase, the final per-label score is given by:

$$P_l = \mathcal{A}(\Phi_l(\mathbf{z}_l^1), \ \Phi_g(\mathbf{z}_l^2)) \cdot \Phi_q(x)$$

**Phase II:** In this phase, we first refine our clusters based on the jointly learnt word embeddings. Specifically, we recluster the labels using the dense $\mathbf{z}_l^1$ representations instead of using their sparse PIFA representations Chang et al. (2020) and consequently reinitialize $\mathcal{W}_m$. We repeat the Phase I training, but this time the formulation of $\mathcal{W}_e$ also includes $\mathbf{w}_l$ which are initialised with the updated $\mathbf{z}_l^1$ as well. Here, the final per-label score is given by:

$$P_l = \mathcal{A}(\Phi_l(\mathbf{z}_l^1), \ \Phi_g(\mathbf{z}_l^2), \ \mathbf{w}_l) \cdot \Phi_q(x)$$

# B  ADDITIONAL VISUALIZATIONS

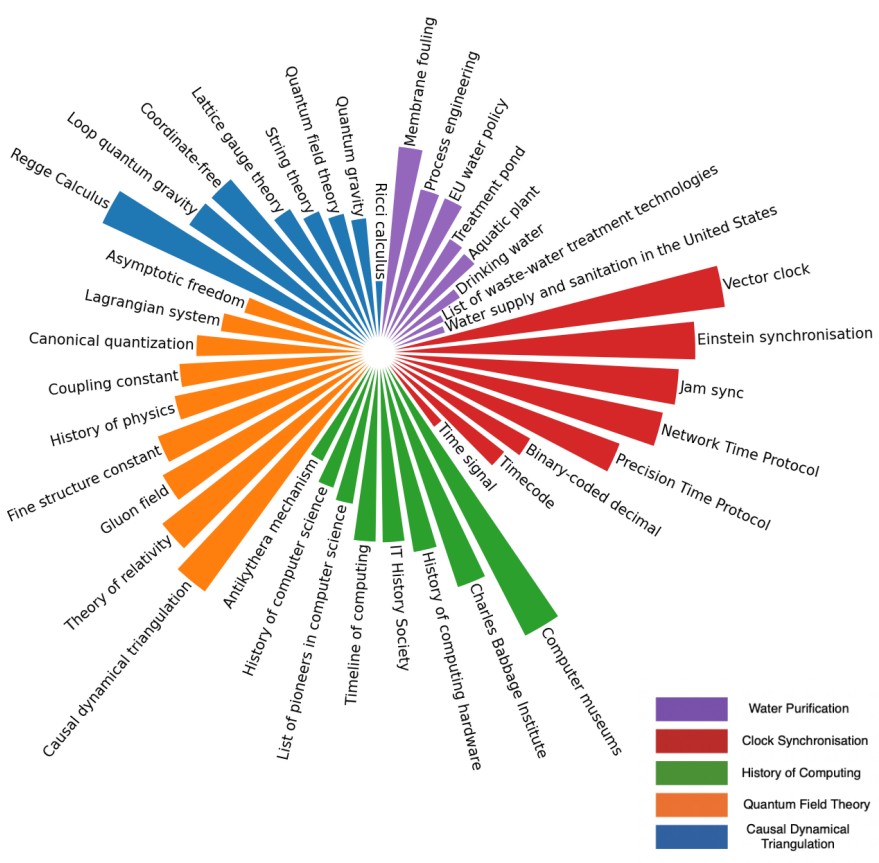

Figure 4: Correlations between labels and their first-order neighbours, as found by the LCG on the LF-WikiTitles-500K dataset. The legend shows the label in question, the bar chart shows the degree of correlation with its neighbouring labels. Correlated labels often share tokens with each other and/or may be used in the same context.

| Method | Datapoint | Baseline Predictions | *Gandalf* Predictions |
|---|---|---|---|
| INCEPTIONXML-LF | Grand Lake, Colorado | Colorado metropolitan areas, Front Range Urban Corridor, **Outline of Colorado, Index of Colorado-related articles**, State of Colorado | Colorado metropolitan areas, **Outline of Colorado, Index of Colorado-related articles, Colorado cities and towns, Colorado counties** |
| DECAF | | Colorado metropolitan areas, Front Range Urban Corridor, State of Colorado, Colorado municipalities, National Register of Historic Places listings in Grand County, Colorado | **Outline of Colorado**, State of Colorado, **Colorado cities and towns**, Colorado municipalities, **Colorado counties** |
| ECLARE | | State of Colorado, **Colorado cities and towns, Colorado counties**, National Register of Historic Places listings in Grand County, Colorado, Grand County, Colorado | **Outline of Colorado, Index of Colorado-related articles**, State of Colorado, **Colorado cities and towns, Colorado counties** |
| INCEPTIONXML-LF | Armed Forces of Saudi Arabia | **Royal Saudi Air Defense, Royal Saudi Strategic Missile Force**, Saudi Royal Guard Regiment, Terrorism in Saudi Arabia, Capital punishment in Saudi Arabia | **Military of Saudi Arabia, Royal Saudi Air Force, Royal Saudi Air Defense, Royal Saudi Strategic Missile Force, King Khalid Military City** |
| DECAF | | Saudi Arabian-led intervention in Yemen, Saudi-led intervention in Bahrain, Human rights in Saudi Arabia, Legal system of Saudi Arabia, Joint Chiefs of Staff (Saudi Arabia) | **Royal Saudi Air Force, Royal Saudi Navy, Royal Saudi Air Defense, Royal Saudi Strategic Missile Force, Saudi Arabian National Guard** |
| ECLARE | | List of armed groups in the Syrian Civil War, **Military of Saudi Arabia, Royal Saudi Strategic Missile Force, King Khalid Military City**, Joint Chiefs of Staff (Saudi Arabia) | **Military of Saudi Arabia, Royal Saudi Air Defense, Royal Saudi Strategic Missile Force, King Khalid Military City**, Saudi Royal Guard Regiment |

Table 5: Prediction examples of additional datapoints from the LF-WikiSeeAlsoTitles-320K dataset. Labels indicate mispredictions.

## C    COMPUTATIONAL COSTS

*Gandalf*, is a data-centric approach that does not increase the computational cost during inference. While the inclusion of label features - which can often run in the order of millions - as additional data points might seem to increase the computational cost during training, through a series of observations, we show that this is in fact not the case. On the contrary, *Gandalf* can help in reducing the memory footprint while training, enabling researchers to use smaller GPUs, and reallocating their compute budget towards longer training schedules. Secondly, we also study the effect of subsampling the labels used for *Gandalf* to demonstrate how learning even some of the label-label correlations is beneficial for XMC models. This observation is particularly useful when inclusion of all label-features as data points becomes intractable due to its scale.

### C.1    COMPUTATIONAL COSTS DURING TRAINING

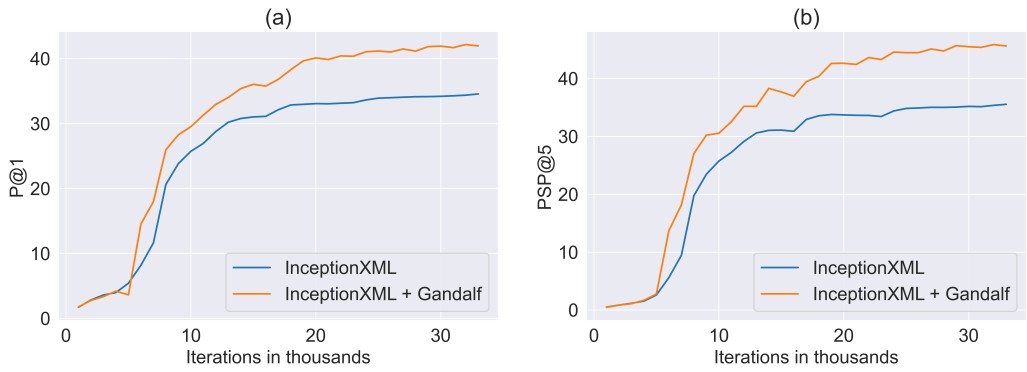

Figure 5: The (a) P@1 and (b) PSP@5 metric for LF-AmazonTitles-131K dataset plotted against iterations for InceptionXML with and without Gandalf.

For the LF-AmazonTitles-131K dataset, we plot the P@1 and the PSP@5 metric against iterations for InceptionXML, trained with and without Gandalf in Figure 5. As can be seen, using Gandalf gives better performance, even on tail labels, right from the beginning. Moreover, where the performance of InceptionXML saturates, the performance of Gandalf continues to scale with increasing compute. Therefore, given a fixed computational budget, a model trained with Gandalf will outperform one trained without it.

Furthermore, the inclusion of label-features in previous XMC works involve architectural additions on base models Mittal et al. (2021a;b). For instance, DECAF Mittal et al. (2021a) involves two forward passes of the Astec base encoder Dahiya et al. (2021b) to include label features in the classifier. On top of this, ECLARE Mittal et al. (2021b) adds a heavy linear layer to learn GALE features. These additions increase the memory footprint by 2x and 3x, respectively. Similarly, InceptionXML-LF is built upon InceptionXML Kharbanda et al. (2023) and increases the memory footprint by 3x. However, it is evident from Table 2 that "Astec + *Gandalf*" outperforms "DECAF/ECLARE + *Gandalf*"[2]. Therefore, we can conclude that *Gandalf* enables us to use lighter architectures, without sacrificing performance.

These observations firmly place *Gandalf* as a compute-efficient method of leveraging label-features in XMC models.[3]

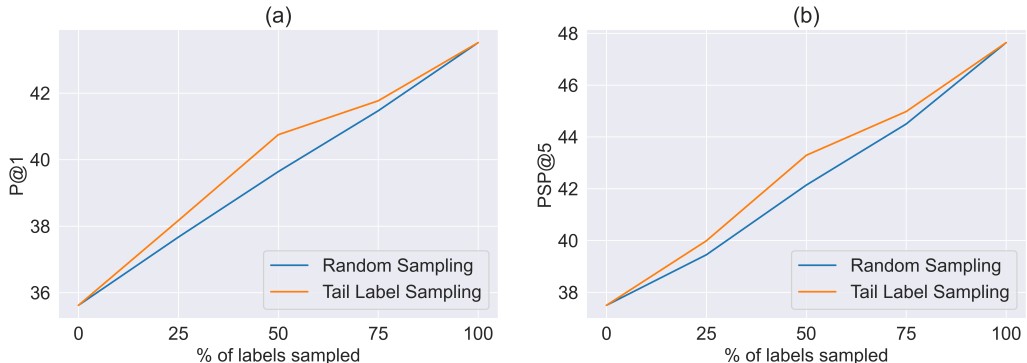

Figure 6: The effect of subsampling labels for *Gandalf* on the (a) P@1 and (b) PSP@5 metric for LF-AmazonTitles-131K dataset.

## C.2 EFFECT OF SUBSAMPLING LABELS

We demonstrate the effect of subsampling labels used for *Gandalf* under two schemes, (a) Randomly sampling an expected percentage subset of labels and (b) randomly sampling this subset from equi-voluminous bins of increasing label frequency, i.e., prioritising tail labels for lower percentages. These results are shown for the P@1 and PSP@5 metric on the LF-AmazonTitles-131K dataset in Figure 6.

Both the metrics grow linearly as the percentage sampled labels are increased in steps of 25%. This goes ahead to show the lack of label-label correlations being captured in existing methods, and how learning even on a subset can be useful. Further, prioritising tail-labels consistently outperforms the random sampling baseline, underscoring the data-scarcity issue in XMC.

---

[2]Note that these algorithms individually also benefit from using *Gandalf*

[3]Note that the creation of the LxL label correlation graph takes less than two minutes, even for the large LF-AmazonTitles-1.3M dataset. This is only done once before training and has a negligible effect on the computational cost.