# OpenReview forum: "Gandalf: Learning label correlations in Extreme Multi-label Classification via Label Features"
_ICLR.cc/2024/Conference — Submitted to ICLR 2024_

### Official Review · Reviewer_pzXY · 2023-11-01

**Soundness:** 3 good
**Presentation:** 2 fair
**Contribution:** 2 fair
**Rating:** 5
**Confidence:** 4

**Summary:**

Gandalf, a novel approach which makes use of a label correlation graph to leverage label features as additional data points to supplement the training distribution. Their approach can be applied in a plug-and-play manner with several existing methodologies, leading to a 30% performance improvement. The authors focus on the short-text setting where they exploit the symmetry between inputs and labels to obtain improved learning of label correlations. They propose an approach leverages the innate symmetry of short-text XMC along the LCG to construct valid data-points.

**Strengths:**

* The approach can be applied in a plug-and-play manner with several existing methodologies - the results of which have been demonstrated in Table 2.
* They have used publicly available benchmark datasets to evaluate their results.
* Gandalf shows relatively large improvement of 30% over 5 state-of-the-art algorithms across 4 benchmark datasets.

**Weaknesses:**

* The authors focus on short text. While it is widely used across the industry, it will be good to demonstrate why their approach is better for short text when compared to other approaches. In a sense what makes the approach more suited for short-text? At the same time, would the approach work on large text as well?

* It will be great if the authors can elaborate on the "Symmetric nature of short-text XMC". An example to illustrate the symmetric form would strengthen the reasoning for utilizing the symmetry. Especially when the paper in a way hinges on the utilizing the symmetry along with the label correlation graph; LCGs have been used in other approaches.
As a result, the novelty seems limited.

*Along the same lines, it will greatly help if the authors can detail how the approach can handle the sparse instance-to-label mapping present in the datasets.


A minor issue: I believe there are a few missing references in the paper and the supplemental material.

**Questions:**

Addressed in the weakness.

---

> ### Author Response · Authors · 2023-11-20
>
> Thank you for the review. Each argument is considered below:
>
>  > The authors focus on short text. While it is widely used across the industry, it will be good to demonstrate why their approach is better for short text when compared to other approaches. In a sense what makes the approach more suited for short-text? At the same time, would the approach work on large text as well?
>
> The approach is particularly well suited for short input text problems such as those encountered in prediction of Related Searches on a search engine or suggestion of similar product titles in response to a user’s purchase/interest of a product on a recommender system. Apart from both the inputs (by the user) and the prediction/recommendation (by a machine learning system) being of short-text nature, these come from the same space. One could equally well replace a prediction by the ML system as an input by a user, making them interchangeable. This is no longer true in long-text problems such as that of assigning tags (with few words) to Wikipedia articles with hundreds of input words.
>
> > It will be great if the authors can elaborate on the "Symmetric nature of short-text XMC". An example to illustrate the symmetric form would strengthen the reasoning for utilizing the symmetry. Especially when the paper in a way hinges on the utilizing the symmetry along with the label correlation graph; LCGs have been used in other approaches. As a result, the novelty seems limited.
>
> Two examples highlighting the symmetric nature of the input and outputs in the short-text XMC problems have already been provided in the Preliminaries section of our paper. We would like to clarify that “symmetry” in the paper refers to the interchangeability of the input data points and the label features, and NOT the fact that “matrix representing the label correlation graph is symmetric”. It is important to remark that the novelty of our approach stems from exploiting the problem structure to propose an efficient and effective data augmentation technique leading to significant performance improvements in prediction accuracy, and not from the choice of LCG.
>
> > *Along the same lines, it will greatly help if the authors can detail how the approach can handle the sparse instance-to-label mapping present in the datasets.
>
> We would like to highlight that all XMC methods handle the sparsity issue by leveraging one-vs-all classifiers. While some approaches directly employ these classifiers [DiSMEC, XR-Linear], the others perform curriculum learning where they first train on a surrogate task where the model is trained on relatively denser instance-to-meta-label mappings by clustering labels into meta labels. Many approaches have also successfully leveraged label-clusters in different ways [AttentionXML, XR-Transformer, CascadeXML]. We do not propose a new approach to tackle this issue. Rather, the proposed Gandalf augmentation enables existing XMC models to learn label-label correlations from this sparse mapping that were previously being missed.

---

### Official Review · Reviewer_tePQ · 2023-11-01

**Soundness:** 3 good
**Presentation:** 2 fair
**Contribution:** 2 fair
**Rating:** 5
**Confidence:** 4

**Summary:**

The paper presents a data augmentation method called Gandalf which leverage label correlation as additional data points against short-text extreme multi-label text classification problem. The presented experiment results show Gandalf is able to improve the performance on other extreme classifiers on several benchmark datasets.

**Strengths:**

1)	The proposed data augumentation idea is relatively simple and work effectively on several benchmark datasets.

2)	The empirical studies are relatively abundant.

**Weaknesses:**

1) The underlying idea of the main method a little bit lacks novelty and seems an extension of the existing work likes ECLARE.

2) The method does not contribute to the real-world settings as most XMC methods choose to make partial experiments on long-text benchmark datasets. Besides, the method seems to increases the overhead of training datasets which may cause limitations.

**Questions:**

1)	Why classical XMC problems like AttetionXML, SiameseXML++ are not experimented with Gandalf?

2)	It seems that the proposed Gandalf does not give competitative performance on PSP metrics compared to existing methods. Do you think Gandalf is an effective method dealing with the tail labels in XMC problem?

3)	Can Gandalf work on long-text XMC datasets?

---

> ### Author Response · Authors · 2023-11-20
>
> Thank you for your constructive comments. We rebut each point below:
>
> > The underlying idea of the main method a little bit lacks novelty and seems an extension of the existing work likes ECLARE.
>
> To the best of our knowledge, this is the first method which proposes a generalisable data augmentation in XMC that leads to significant improvements over a variety of baselines methods. ECLARE is only related to our approach in that their proposed label correlation graph is one of the possible ways to annotate label features. This is merely an implementation detail, and our novelty lies in the way we annotate label features with label correlations.
>
> More specifically, ECLARE uses the LCG to generate an additional feature "graph-augmented label embedding" i.e. GALE, whereas Gandalf uses the LCG to get the supervised classifier targets for label features. This has also been mentioned in “Gandalf vs ECLARE” under section 5.2 of the revised submission.
>
> > The method does not contribute to the real-world settings as most XMC methods choose to make partial experiments on long-text benchmark datasets. Besides, the method seems to increase the overhead of training datasets which may cause limitations.
>
> We would like to reiterate that short-text XMC has a very high impact in the real-world, as also acknowledged by Reviewer pzXY. It forms the backbone for search queries on leading search engines and product recommendation systems and for ad-phrase matching [DECAF, ECLARE, SiameseXML, NGAME]. For computational overheads, please refer to the common rebuttal.
>
> > Why classical XMC problems like AttetionXML, SiameseXML++ are not experimented with Gandalf?
>
> For short-text problems, our method is also applicable to AttentionXML, which has been superseded by many recent state-of-the-art methods. SiameseXML does not use traditional XMC losses(Binary Cross-Entropy Loss, Hinge Loss, Squared Hinge Loss). Instead, it uses a customized loss named “kProbContrastive” in code and explained in the paper. Our method is suitable for traditional XMC losses, and we show that using Gandalf is sufficient to outperform such especially designed loss functions.
>
> Through our experiments, we find that modifying Astec with Gandalf is more beneficial than modifying it with SiameseXML training schedules and loss functions (Note that Astec and SiameseXML use the same architecture).
>
> > It seems that the proposed Gandalf does not give competitative performance on PSP metrics compared to existing methods. Do you think Gandalf is an effective method dealing with the tail labels in XMC problem?
>
> Contrary to the reviewer’s assertion that Gandalf does not improve upon PSP metrics, we would like to bring to attention Table 2, where Gandalf consistently increases the outperforms the unaugmented models, with absolute improvements up to 10 additional points on the PSP metrics. This shows its  effectiveness in dealing with tail labels.
>
> This has further been shown in Figure 2, where we show the P@5 scores with consistent improvements, particularly on head and tail labels.
>
> > Can Gandalf work on long-text XMC datasets?
>
> Gandalf is rooted in the fact that the input data points and label features are symmetric in nature. Paraphrasing our arguments from “Symmetric nature of short-text XMC with Label Features” in the manuscript, this symmetry arises from both of them being short-text features. Since this does not hold for long-text datasets, as also mentioned in the “Conclusion”, it does not directly apply for such scenarios.
>
> Data augmentation for long-text problems could potentially involve using an external source such as an LLM to generate similar passages to input long-text data points without changing the label annotations. Unlike Gandalf this could be much more computationally expensive and beyond the scope of our paper.

---

### Official Review · Reviewer_Wc5R · 2023-11-02

**Soundness:** 3 good
**Presentation:** 3 good
**Contribution:** 2 fair
**Rating:** 6
**Confidence:** 5

**Summary:**

The paper proposes Gandalf, which augments the training dataset in extreme classification by using label features as documents, with their corresponding “label mapping” being constructed using a label-label graph. Such a setup allows most existing extreme classifiers to now leverage label features for improved generalization, without any changes to the training pipeline and no added inference cost. Gandalf shows significant improvement in both Precision and Propensity-scored Precision metrics over four commonly used extreme classification datasets.

**Strengths:**

- The proposed methodology is architecture-agnostic thereby resulting in easy and widespread adoption.
- Consistently improved performance for a variety of extreme classifiers, especially for tail labels.

**Weaknesses:**

- The training time will be significantly increased since the new number of training points will be number of documents + number of labels. And in typical extreme classification setups, the number of labels can be much greater than the number of available documents.
- The approach assumes (1) label features exist in the same input space as documents; and (2) the extreme classifier is NOT a two-tower approach, and embeds the labels and documents in the same space.

**Questions:**

- What if you use graphs other than the random-walk graph in ECLARE? For example, the co-occurrence graph?
- Why should the performance of classifiers that already use label-features (e.g., ECLARE, DECAF) improve with Gandalf?
- To combat the increased training cost, it would be interesting to understand the sample efficiency of Gandalf. To be more specific, how much performance is improved when augmenting, e.g., {0, 25, 50, 75, 100}% of random labels to the training data?
- Missing citations in multiple places in the paper.

---

> ### Author Response · Authors · 2023-11-20
>
> Thank you for the review. Each argument is considered below:
>
> > The training time will be significantly increased since the new number of training points will be number of documents + number of labels. And in typical extreme classification setups, the number of labels can be much greater than the number of available documents.
>
> This has been answered in the common rebuttal, and added to Appendix C of the revised submission.
>
> > The approach assumes (1) label features exist in the same input space as documents; and (2) the extreme classifier is NOT a two-tower approach, and embeds the labels and documents in the same space.
>
> We believe there has been a misunderstanding and the below two points have been confused:
> * The idea that label features co-exist in the same distribution as input queries is a well-accepted and proven fact in short-text XMC that has been leveraged in former two-tower XMC approaches like SiameseXML and NGAME.
> * Two-tower approaches predominantly only make use of an encoder and are trained for retrieval tasks. By extreme classifiers we essentially imply encoders with an additional large classifier component i.e. the last classification layer having the number of weight vectors equal to the number of labels.
> Please check “Two-tower Models & Classifier Learning '' under “Related Work” for further discussion.
>
> > What if you use graphs other than the random-walk graph in ECLARE? For example, the co-occurrence graph?
>
> We tried this experiment and provide scores on InceptionXML below. While this also leads to improvements over baselines, it still does not surpass the gains obtained from LCG (ECLARE). We posit that this is due to extra correlations being captured via random walk in the latter.
> * LF-AmazonTitles-131K: P@1: 41.39, P@3: 26.93, P@5: 19.41, PSP@1: 34.38, PSP@3: 40.81, PSP@5: 45.13
> * LF-WIkiSeeAlsoTitles-320K: P@1: 29.64, P@3: 19.78, P@5: 15.04, PSP@1: 22.81, PSP@3: 25.18, PSP@5: 26.96
>
> > Why should the performance of classifiers that already use label-features (e.g., ECLARE, DECAF) improve with Gandalf?
>
> Classifiers like ECLARE and DECAF incorporate architectural strategies in order to capture higher order document-label correlations. In this work, we aim to learn label-label correlations which were not captured by previous models. Gandalf is designed to explicitly capture label-label correlations, which leads to significant improvements. This has also been discussed in Section 5.2, “Gandalf vs ECLARE”.
> To combat the increased training cost, it would be interesting to understand the sample efficiency of Gandalf. To be more specific, how much performance is improved when augmenting, e.g., {0, 25, 50, 75, 100}% of random labels to the training data?
> That is an interesting idea! We tried this and added the results in Appendix C.2. Summarizing:
> * As suggested, we randomly sample {0, 25, 50, 75, 100}% of the labels and augment the training dataset with their annotations from Gandalf. With random sampling, we note a near linear increase in scores with an increasing number of labels in the dataset. This signifies the importance of learning label-label correlations, and the efficacy of Gandalf towards this goal.
> * Moreover, we also perform this sampling while prioritizing tail labels. More specifically, we sample {0, 25, 50, 75, 100}% of the labels from equi-voluminous bins of labels of increasing frequency. This implies that the added labels would primarily be tail labels for lower sampling percentages. We note that this consistently outperforms the random sampling baselines, with improvements even on the PSP@5 metric. This experiment shows that Gandalf is also useful in tackling the data-scarcity issue in XMC.

---

> > ### Comment · Reviewer_Wc5R · 2023-11-23
> > **Response to authors**
> >
> > Thank you for your response and additional experiments. I would like to stick to my original rating.

---

### Official Review · Reviewer_S9V4 · 2023-11-05

**Soundness:** 2 fair
**Presentation:** 3 good
**Contribution:** 2 fair
**Rating:** 5
**Confidence:** 4

**Summary:**

This paper studies Extreme Multi-label Text Classification (XMC) problems, which assigns a short text input sample with a subset of most relevant labels from millions of label choices. The principal difficulty in XMC is managing the vast array of possible classes. Building upon existing research, this work incorporates the textual features of labels into the classifier's training process. Especially, given that input samples often share common tokens with the labels they're associated with, this task becomes correlating short text inputs with related sets of text.  For example, the input sample “2022 French presidential election” could be associated with “April 2022 events in France”,  “2022 French presidential election”, “2022 elections in France”, “Presidential elections in France.”

While previous research has explored various methods for aligning input and label texts, this paper proposes a straightforward technique for data augmentation, illustrated in Figure 1. It enhances the original N*L training data matrix with an additional L*L matrix, which captures the interrelationships between the L labels. The results from experiments suggest that this enrichment with the L*L matrix enables established XMC classifiers to attain better accuracy in classification tasks.

**Strengths:**

1)	The data augmentation concept introduced in this paper is refreshingly straightforward, offering an intuitive strategy to expand the training dataset.
2)	Empirical assessments indicate that incorporating this additional data into the training process proves beneficial.

**Weaknesses:**

1.	The augmented dataset introduced is considerably large, e.g., potentially consisting of a large matrix in size of millions by millions. The training time will be significantly increased.  While it's true that this does not affect the inference time, it substantially extends the duration of the training phase due to the increased volume of data.
2.	The two-tower model “NGAME + classifier” yields the highest performance on the Amazon datasets. Even with the introduction of additional data, the base algorithms do not surpass the efficacy of the two-tower.
3.	Some reference citations are missing: Zhang et al., 2021a; ?; Lu et al., 2022),  ANCE (?)

**Questions:**

1)	What is the computation cost for training the base algorithms when taking the additional L*L matrix? including the process of obtaining the L*L matrix.  It would be beneficial for the paper to detail the expected impact on the training duration.
2)	Why there is no evaluation of two-tower models on the two wiki-dataset?

---

> ### Author Response · Authors · 2023-11-20
>
> Thank you for the review. Each argument is considered below:
>
> > The augmented dataset introduced is considerably large, e.g., potentially consisting of a large matrix in size of millions by millions. The training time will be significantly increased. While it's true that this does not affect the inference time, it substantially extends the duration of the training phase due to the increased volume of data. & What is the computation cost for training the base algorithms when taking the additional LL matrix?
>
> Please refer to “Additional Training Time and Computational Cost” in the Common Rebuttal
>
> > The two-tower model “NGAME + classifier” yields the highest performance on the Amazon datasets. Even with the introduction of additional data, the base algorithms do not surpass the efficacy of the two-tower.
>
> While the two-tower model,  “NGAME + classifier” yields the highest performance on the Amazon datasets, it must be noted that it uses a 6 layer DistilBERT as its encoder, in comparison to the frugal MLP encoders used in our experiments. We do, in fact, demonstrate the efficacy of Gandalf over two-tower models by comparing with “SiameseXML++ + Classifier”, which proposes a two-tower model on a frugal MLP encoder. While this was mentioned in the manuscript in Table 2, we have made it clearer in the revised submission. Reiterating, all baselines above the double line denote transformer based encoders, and below it denote MLP encoders. The * denotes two-tower approaches, as mentioned in the caption.
>
> > Why there is no evaluation of two-tower models on the two wiki-dataset?
>
> As a standard practice in the domain, recent papers only evaluated the performance of “Dense Retrieval Two Tower” approaches on Amazon Datasets.
> The two-tower model scores have been taken from previous publications which have not reported their scores for the wiki-datasets. Provided that it is non-trivial to run DR models directly on XMC datasets and due to our restricted computational budget, we could not run the heavy DR models on these datasets ourselves.

---

### Author Response · Authors · 2023-11-20
**Common Rebuttal: Additional training time and computation cost**

We would like to thank the reviewers for their time and constructive comments. Below, we provide a section to answer questions common to multiple reviewers. Where applicable, we will refer to this as the “common rebuttal”. Reviewer specific comments have been responded to individually. Where relevant, we have also referred to the revised submission (also corrected missing references in the same). Looking forward to a fruitful discussion!

We have provided a detailed explanation of the computational cost in Appendix C of the revised submission. Below, we summarize the key highlights:
* We would like to highlight that while the addition of extra data points increases the training time by 1.2 - 1.5x (depending on the dataset), this gives a large relative  performance improvement of over 30% over its baseline.
* This improvement comes with no additional computational cost during inference, as compared to DECAF and ECLARE, which make architectural additions to incorporate label features. This leads to a 2 - 3x increase in both training and inference time.
* A model trained with Gandalf outperforms one trained without it, when trained for the same number of iterations, i.e, for equal training times. Moreover, a model trained without Gandalf saturates earlier, while the performance of Gandalf continues to scale positively with longer training schedules.
* Creation of the LxL matrix takes under two minutes, even for the largest dataset, LF-AmazonTitles-1.3M. This is a negligible overhead before training and does not add any additional cost per iteration. Moreover, all the data points are created on the fly and thus do not need to be stored on the disk.

---

### Meta-Review · Area_Chair_Wk5i · 2023-12-06

**Metareview:**

This paper proposes a data augmentation method named Gandalf to address the extreme multi-label text classification problem. In particular, the proposed method exploits label correlation as additional data points. The paper is well organized and clearly written. Experiments demonstrate the effectiveness of Gandalf in most cases. However, reviewers raised many concerns, such as the additional computational costs during training, insufficient evaluations (e.g., SiameseXML++ with Gandalf), and potential extension to long-text XMC. Unfortunately, the authors' responses cannot fully address these concerns. Therefore, this paper in its current version is not ready for publication at ICLR.

**Justification For Why Not Higher Score:**

Several major limitations are not clearly addressed, such as the training cost and evaluation of SiameseXML++ with Gandalf.

**Justification For Why Not Lower Score:**

N/A

---

### Decision · Program_Chairs · 2024-01-16

Reject